# Deliberate practice for retinopathy of prematurity: Retinal laser training using schematic eyes in ophthalmology education

Narisa Rattanalert[1]*, Supaporn Tengtrisorn[1], Phanthipha Wongwai[2], Atchareeya Wiwatwongwana[3], Penny Singha[1], Sirinya Suwannaraj[2], Thunyaluck Jiwanarom[3], Warachaya Phanphruk[2], Parichat Damthongsuk[1], Dorene F. Balmer[4]

1 Department of Ophthalmology, Faculty of Medicine, Prince of Songkla University, Hat Yai, Songkhla, Thailand, 2 Department of Ophthalmology, Faculty of Medicine, Khon Kaen University, Khon Kaen, Thailand, 3 Department of Ophthalmology, Faculty of Medicine, Chiang Mai University, Chiang Mai, Thailand, 4 The Children's Hospital of Philadelphia, University of Pennsylvania School of Medicine, Philadelphia, Pennsylvania, United States of America

* narisa.r@psu.ac.th (NR)

## Abstract

Skills and confidence in performing high-risk procedures are essential for ensuring safe patient care. Deliberate practice is an instructional model designed to improve performance by engaging trainees in structured, repetitive practice with immediate feedback provided by supervisors. We developed a multifaceted simulation informed by deliberate practice and tested the hypothesis that trainee skills and reported confidence would increase after training. A multicenter prospective study was conducted at three universities in Thailand from July 1, 2023, to June 30, 2024. Sixty ophthalmology trainees participated in an introductory video for self-study and then completed a multiple-choice questionnaire to assess their baseline knowledge of laser indirect ophthalmoscopy for retinopathy of prematurity (LIO-ROP) and guide feedback by supervisors. The participants subsequently performed a simulated LIO-ROP on a schematic eye and received additional feedback based on a scoring rubric. The participants practiced on the schematic eye as much as needed to feel confident in their skills. Mean rubric scores indicative of LIO-ROP skills significantly improved from 2.94 to 3.59 out of 4 (P < 0.001), and the time required for the laser procedure decreased from 17.19 to 15.14 minutes in pre- and post-LIO-ROP practice, respectively. Rubric scores for performing the LIO-ROP on a schematic eye significantly improved across all steps of the procedure between pre- and post-LIO-ROP practice. Reported confidence in performing the LIO-ROP increased by 81.5%. Multifaceted simulated training informed by deliberate practice is a suitable instructional model for enhancing skill performance and confidence among postgraduate residents.

**Data availability statement:** All relevant data are within the paper and its Supporting Information files.

**Funding:** This study was funded by the Collaborative Research Grants of CMU-KKU-PSU Cooperation Project by Faculty of Medicine of Chiang Mai University in the form of a grant [021/2567 to AW], the Khon Kaen University in the form of a grant [67/1-2-2-2-001 to PW], and the Prince of Songkla Univerisity in the form of a grant [65/1-PSU-2-1-1 to ST]. The funders had no role in study design, data collection and analysis, decision to publish, or preparation of the manuscript.

**Competing interests:** The authors have declared that no competing interests exist.

## Introduction

Skills and confidence in performing high-risk procedures are essential for safe patient care. Ideally, trainees should have sufficient time and low-risk learning opportunities to build their skills and confidence before progressing to on-the-job learning. Deliberate practice [1] is an instructional model that optimizes learning by engaging trainees in well-defined tasks at appropriate levels of difficulty under the supervision of teachers who provide formative feedback. This approach involves repetitive practice and self-monitoring, ultimately fostering mastery, enhanced skill, and improved patient safety.

Deliberate practice has been widely applied in fields such as chess, music, typing, sports, and postgraduate medical education [2,3]. For example, it has been used to teach life-saving procedures, such as cricothyroidotomy, to emergency medicine residents [4].

In pediatric ophthalmology, laser indirect ophthalmoscopy for the treatment of retinopathy of prematurity (LIO-ROP) requires a high level of knowledge and skill. Ophthalmology residents and fellows in Thailand currently receive on-the-job training at university hospitals. Postgraduate training typically begins with knowledge acquisition and progresses to hands-on practice with the patients under supervision. However, this on-the-job training is associated with substantial risks to patients, such as incorrect laser direction, complications like cataracts or macular scars, and prolonged general anesthesia due to extended procedure times. These challenges and the inherent difficulties of on-the-job training could negatively affect learning if trainees feel overwhelmed by emotional stress or lose confidence after making procedural errors [5,6].

Studies across various contexts and learner groups [7–12] demonstrate that training in simulated environments can improve knowledge, skills, and confidence in surgical procedures. While simulations would ideally be incorporated into pediatric ophthalmology training before practicing laser indirect ophthalmoscopy (LIO) on actual patients [13–17], such resources are not currently available in Thailand's training programs. To address this gap, we developed a multifaceted deliberate practice module using an LIO-ROP simulator.

This study's purpose was to evaluate the effectiveness of a multifaceted deliberate practice-based training module for LIO-ROP using schematic eyes and assess its impact on trainee skills and confidence.

## Methods

A multicenter prospective study was conducted in the Department of Ophthalmology at three universities in Thailand from July 1, 2023, to June 30, 2024. All procedures involving human participants were conducted in accordance with the ethical standards of the Human Research Ethics Committee of the Faculty of Medicine at Prince of Songkla University (REC.66-216-2-1), Khon Kaen University (HE661350), and Chiang Mai University (OPT-2566–0330). The participants were residents and fellows from the ophthalmology departments of three universities. All participants were verbally informed by the investigators, in the absence of witnesses, before signing the consent form. They were provided with a hard copy of the consent form, which included comprehensive details of the study, as well as an emergency contact number.

## Educational intervention

The multifaceted deliberate practice module comprised the self-study LIO-ROP video (VDO), multiple-choice questions (MCQ) for baseline knowledge assessment, gap knowledge feedback, pre-practice LIO-ROP using the schematic eye, skill assessment based on a rubric score and supervisor feedback, self-practice using the schematic eye, post-practice LIO-ROP on the schematic eye, and skill assessment with rubric scoring by the supervisor. A Usability Experience Questionnaire (UEQ) was used for product evaluation, along with a self-confidence assessment.

## Learning scheme

Sixty participants from three universities (20 per university) provided informed consent and shared their demographic information before beginning the self-study VDO session. Baseline knowledge of the LIO-ROP steps was assessed using an MCQ test, followed by feedback and clarification of correct answers from supervisors. Pre-practice sessions on the schematic eye included formative feedback on participants' techniques, based on rubric scores (described below). Participants then engaged in self-practice on the schematic eye until they demonstrated proficiency and felt confident enough in their own skills for the next step (Fig 1).

## VDO and schematic eye session

An 8-minute VDO session was developed for self-learning, covering the basics and safety of LIO-ROP, treatment steps, instrument setup, laser settings, and serious complications. A portable schematic eye was created using 3D printing at a reasonable cost of approximately 130 US dollars. As shown in Fig 2, a fundus photograph of the ROP was printed and placed on carbon paper to simulate the laser treatment area.

## Learning assessment

**Baseline knowledge assessment.** An 18-item MCQ was developed based on content requirements corresponding to the VDO (S1 file). Content experts verified the questions, which covered LIO safety, treatment steps, essential instruments, laser settings, and precautions for complications.

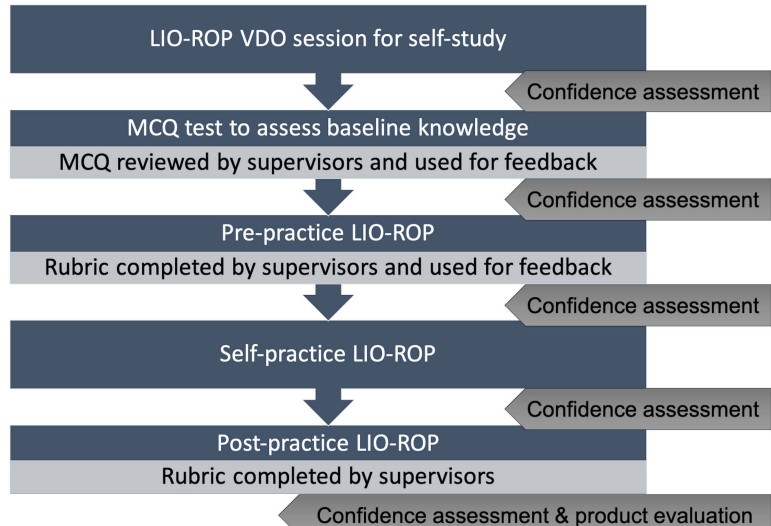

**Fig 1. The study flow for participants.** Abbreviations: LIO-ROP, laser indirect ophthalmoscopy for retinopathy of prematurity; VDO, Video; MCQ, multiple-choice question.

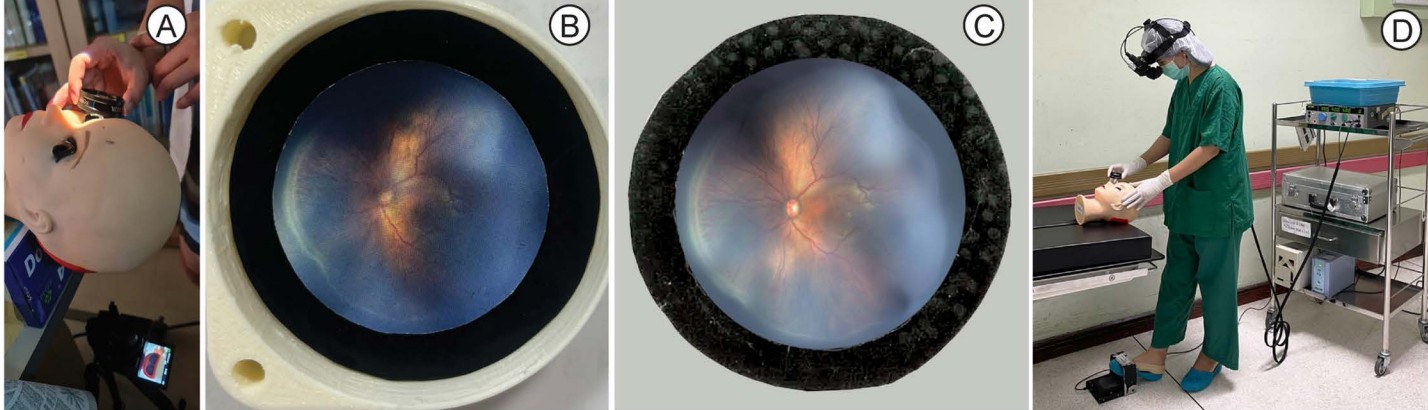

**Fig 2. Schematic eye for LIO-ROP training and practice.** (A) Model for LIO-ROP training, (B) Fundus photograph without laser reaction on carbon paper, (C) Fundus photograph with laser reaction on carbon paper, (D) LIO-ROP performed on schematic eye. Abbreviation: LIO-ROP, laser indirect ophthalmoscopy for retinopathy of prematurity.

**Skill assessment.** The LIO-ROP procedure was reviewed by pediatric ophthalmologists, and its steps were summarized by consensus. A scoring rubric was developed, modified from the ICO Ophthalmology Surgical Competency Assessment Rubric (ICO-OSCAR) for pan-retinal photocoagulation [18], a standardized tool for evaluating office-based laser procedures for retinal diseases (S2 file). The LIO-ROP skill assessment included 13 treatment steps and five global indices. The performance levels for each step were graded using the modified Dreyfus model as follows: 1 = novice, 2 = beginner, 3 = advanced beginner, and 4 = competent [19]. Assessments were administered before and after LIO practice to measure improvements in surgical skills.

**Confidence and user experience assessment.** Participants evaluated the schematic eye model using the Thai version of the Usability Experience Questionnaire (UEQ) [20], which includes dimensions such as attractiveness, perspicuity, efficiency, dependability, stimulation, and novelty. Written feedback on the schematic eye model, training flow, and overall experience was also collected (Fig 1, S3 file).

## Statistical analysis

The sample size was calculated [21,22], to detect differences in evaluation rubric scores between the pre- and post-training LIO-ROP sessions on the schematic eye. Based on a power of (beta) 0.8 and an alpha of 0.05, 32 participants were required to complete the module.

Continuous variables were analyzed and presented as means ± standard deviations (SD). Discrete variables were expressed as proportions (%) to illustrate the relative frequencies of the different categories. Paired t-tests were used to assess changes in continuous data before and after the intervention. The UEQ was used to collect detailed feedback on the schematic of the eye model. The internal consistency of the rubric scores and UEQ was evaluated using Cronbach's alpha.

## Results

Participant demographics are presented in Table 1. Of the 60 participants, 35 (58%) had prior experience with laser application in patients, and eight (13%) had prior experience with LIO in patients with ROP (Table 1).

The mean ± SD MCQ score after the LIO-ROP VDO session was 15.05 ± 1.57 out of 18. The LIO-ROP rubric scores significantly improved between pre- and post-practice in all steps, indicating increased trainee skills. The mean ± SD

**Table 1. Demographic data.**

| Baseline characteristics | n (%) |
|---|---|
| Sex | |
| Male | 18 (30.0) |
| Female | 42 (70.0) |
| Training program | |
| Resident | 56 (93.3) |
| Fellowship | 4 (6.7) |
| Training years and status | |
| Resident year 1 | 12 (20.0) |
| Resident year 2 | 18 (30.0) |
| Resident year 3 | 19 (31.7) |
| Resident year 4 | 7 (11.7) |
| Fellowship | 4 (6.6) |
| Retinal laser treatment experience | |
| No | 25 (41.7) |
| Yes | 35 (58.3) |
| LIO in ROP patient experience | |
| No | 52 (86.7) |
| Yes | 8 (13.3) |

Abbreviations: LIO, laser indirect ophthalmoscopy; ROP, retinopathy of prematurity.

evaluation score for the steps of LIO improved from $3.00 \pm 0.66$ to $3.62 \pm 0.48$ (P<0.001), while the mean±SD evaluation score for global indices improved from $2.89 \pm 0.67$ to $3.57 \pm 0.47$ (P<0.001). The mean time for LIO-ROP decreased from 17.19 minutes to 15.14 minutes (P=0.055).

The rubric scores for performing LIO-ROP on a schematic eye between pre- and post-LIO-ROP practice were significantly improved in all steps (P<0.05), particularly in the steps of laser spot placement, distribution, and adequate laser spot coverage (Table 2). The scores were not significantly different between the experienced and inexperienced groups for any step. Cronbach's alpha for the rubric scores was 0.89 for pre-practice and 0.86 for post-practice.

Regarding user experience, as measured by the UEQ, participants rated the schematic eye above average in six dimensions: attractiveness, perspicuity, efficiency, dependability, stimulation, and novelty, compared with products in the benchmark dataset. The UEQ had a reliability coefficient of 0.9.

Self-reported confidence improved consistently from 56.5% to 81.5% after each step of the process (Table 3). Participants' comments corroborated this increase in confidence in performing the LIO-ROP.

## Discussion

Our multifaceted deliberate practice-based training module for LIO-ROP using schematic eyes showed that mean rubric scores, indicative of LIO-ROP skills, significantly improved in all steps, and the time required for the laser procedure decreased between pre- and post-LIO practice, indicating improved trainee skills. Additionally, trainees reported that their confidence in performing LIO-ROP increased by 81.5%.

Pediatric ophthalmology is a relatively small part of postgraduate medical education. However, as in many other specialties, trainees are required to learn high-risk procedures "on the job," which can compromise patient safety. LIO remains a standard treatment for ROP and is critical for preventing blindness [23,24]. The duration of treatment and the adequacy of spot coverage are crucial to overall treatment success. Our study demonstrates that multifaceted simulated training informed by deliberate practice can considerably decrease treatment time through enhanced skill performance, particularly in the steps of laser spot placement, distribution, and adequate laser spot coverage. These findings have important

**Table 2. Differences in rubric scores between pre- and post- LIO-ROP practice.**

| | Pre-test | Post-test | Diff (Post-Pre) | P Value |
|---|---|---|---|---|
| Total evaluation score | | | | |
| Mean ± SD | 2.94 ± 0.65 | 3.59 ± 0.48 | 0.65 ± .0.49 | < 0.001 |
| Median (IQR) | 2.94 (2.44, 3.5) | 3.75 (3.39, 4) | | < 0.001 |
| **Steps of LIO** | | | | |
| Evaluation score of steps of LIO | | | | |
| Mean ± SD | 3.00 ± 0.66 | 3.62 ± 0.48 | 0.62 ± 0.51 | < 0.001 |
| Median (IQR) | 3 (2.57, 3.46) | 3.79 (3.44, 4) | | < 0.001 |
| Laser radiation safety | | | | |
| Mean ± SD | 3.10 ± 0.61 | 3.69 ± 0.93 | 0.60 ± 0.85 | < 0.001 |
| Median (IQR) | 3 (2,4) | 4 (3.5, 4) | | < 0.001 |
| Safety goggle choice | | | | |
| Mean ± SD | 3.08 ± 1.15 | 3.45 ± 1.02 | 0.37 ± 1.13 | 0.028 |
| Median (IQR) | 3.5 (2.5, 4) | 4 (3.5, 4) | | 0.007 |
| Lens holding | | | | |
| Mean ± SD | 3.09 ± 0.92 | 3.70 ± 0.60 | 0.61 ± 0.83 | < 0.001 |
| Median (IQR) | 3 (2,4) | 4 (4.5, 4) | | < 0.001 |
| Site verification | | | | |
| Mean ± SD | 3.04 ± 0.88 | 3.65 ± 0.65 | 0.61 ± 0.78 | 0.001 |
| Median (IQR) | 3 (3,4) | 4 (3,4) | | 0.001 |
| Dilate pupil | | | | |
| Mean ± SD | 3.60 ± 0.60 | 4 ± 0 | 0.4 ± 0.60 | 0.008 |
| Median (IQR) | 4 (3,4) | 4 (4) | | 0.008 |
| Speculum | | | | |
| Mean ± SD | 3.63 ± 0.60 | 4 ± 0 | 0.37 ± 0.60 | 0.015 |
| Median (IQR) | 4 (3,4) | 4 (4) | | 0.015 |
| Indentation | | | | |
| Mean ± SD | 3.58 ± 0.69 | 4 ± 0 | 0.42 ± 0.69 | 0.016 |
| Median (IQR) | 4 (3,4) | 4 (4) | | 0.015 |
| Laser spot size | | | | |
| Mean ± SD | 3.25 ± 0.80 | 3.69 ± 0.47 | 0.44 ± 0.88 | 0.008 |
| Median (IQR) | 3 (3,4) | 4 (3,4) | | 0.014 |
| Laser setting (Power, exposure, interval) | | | | |
| Mean ± SD | 2.93 ± 0.75 | 3.56 ± 0.57 | 0.63 ± 0.72 | < 0.001 |
| Median (IQR) | 3 (2,3) | 4 (3,4) | | < 0.001 |
| Laser testing and lining | | | | |
| Mean ± SD | 2.89 ± 0.71 | 3.64 ± 0.52 | 0.75 ± 0.75 | < 0.001 |
| Median (IQR) | 3 (2,3) | 4 (3,4) | | < 0.001 |
| Laser spot placement and distribution | | | | |
| Mean ± SD | 2.71 ± 0.78 | 3.41 ± 0.65 | 0.70 ± 0.85 | < 0.001 |
| Median (IQR) | 3 (2,3) | 3 (3,4) | | < 0.001 |
| Adequate laser spot coverage | | | | |
| Mean ± SD | 2.75 ± 0.70 | 3.47 ± 0.63 | 0.73 ± 0.78 | < 0.001 |
| Median (IQR) | 3 (2,3) | 4 (3,4) | | < 0.001 |
| Realize cornea erosion | | | | |
| Mean ± SD | 3.5 ± 0.73 | 4 ± 0 | 0.5 ± 0.73 | 0.015 |
| Median (IQR) | 4 (3,4) | 4 (4) | | 0.015 |

*(Continued)*

**Table 2.** (Continued)

| | Pre-test | Post-test | Diff (Post-Pre) | P Value |
|---|---|---|---|---|
| **Global Indices** | | | | |
| Evaluation score of global indices | | | | |
| Mean±SD | 2.89±0.67 | 3.57±0.47 | 0.68±0.56 | < 0.001 |
| Median (IQR) | 3 (2.4, 3.2) | 3.6 (3.2, 4) | | < 0.001 |
| Knowledge of instrument | | | | |
| Mean±SD | 3.36±0.69 | 3.67±0.65 | 0.31±0.90 | 0.031 |
| Median (IQR) | 3 (3,4) | 4 (3,4) | | 0.028 |
| Flow of the procedure | | | | |
| Mean±SD | 2.74±0.85 | 3.62±0.49 | 0.88±0.70 | < 0.001 |
| Median (IQR) | 3 (2,3) | 4 (3,4) | | < 0.001 |
| Maintaining laser focus | | | | |
| Mean±SD | 2.73±0.81 | 3.56±0.57 | 0.83±0.75 | < 0.001 |
| Median (IQR) | 3 (2,3) | 4 (3,4) | | < 0.001 |
| Continuous laser setting adjustment | | | | |
| Mean±SD | 2.93±0.70 | 3.59±0.59 | 0.66±0.71 | < 0.001 |
| Median (IQR) | 3 (2,3) | 4 (3,4) | | < 0.001 |
| Continuous verification of macula and retinal ridge | | | | |
| Mean±SD | 3.04±0.79 | 3.63±0.56 | 0.59±0.71 | < 0.001 |
| Median (IQR) | 3 (2.5, 4) | 4 (3,4) | | < 0.001 |

Abbreviations: LIO-ROP, laser indirect ophthalmoscopy for retinopathy of prematurity; SD, standard deviation; IQR, interquartile range.

**Table 3. Self-confidence for LIO-ROP after each step of training (score 0–4).**

| Procedure | Score mean±SD | Median (min, max) | % |
|---|---|---|---|
| After the LIO-ROP VDO study | 2.26±0.93 | 2.0 (0, 4) | 56.5 |
| After the MCQ test and comments from supervisors | 2.50±0.82 | 2.5 (1,4) | 62.5 |
| After the first LIO-ROP in the schematic eye and assessment and feedback from the supervisors | 2.97±0.67 | 3.0 (2,4) | 74.3 |
| After practicing LIO-ROP in the schematic eye | 3.00±0.59 | 3.0 (2,4) | 75.0 |
| After the final LIO-ROP in the schematic eye and the assessment | 3.26±0.58 | 3.0 (2,4) | 81.5 |

Abbreviations: VDO, Video; MCQ, multiple-choice question; LIO-ROP, laser indirect ophthalmoscopy for retinopathy of prematurity; SD, standard deviation.

implications for patient care and safety and for other specialties that rely on simulation-based training. The studies by Petrosoniak [4] and Cardoso [9] also showed that deliberate practice and surgical simulation training improve surgical skill.

Our study also showed consistent improvements in trainee confidence with every step of the LIO-ROP training. While competence is often regarded as a higher level of learning than confidence, both are important to consider. Gottleib [25] suggested that although competence receives the greatest attention in postgraduate medical education, confidence must also be addressed, as both are integral to ensuring safe and professional practice [12]. Repetitive surgical simulation training has been shown to improve confidence, which correlates with improved clinical practice [8,9].

Various rubric score models exist for assessing surgical skills, but few specifically address laser skills for ROP [18,26]. Based on a literature review and expert opinions, we developed a rubric to assess LIO-ROP skills. This rubric provided formative feedback that postgraduate doctors could use to improve their performance. Consistent with the principles

of deliberate practice, experienced supervisors provided this feedback in a low-stakes simulation setting, encouraging trainees to use the assessment data for learning and improvement rather than treating it as a high-stakes summative assessment.

We acknowledge the limitations of this study, including the absence of a comparison group to better evaluate changes in LIO-ROP skills. Additionally, the laser reaction and retinal details in the schematic eyes may not fully replicate those in actual patients.

In future studies, researchers should explore the transferability of skills learned in multifaceted, deliberate practice-based simulated environments to real-world clinical practice and evaluate their impact on mastery learning.

In conclusion, multifaceted simulated training informed by deliberate practice represents a suitable model for enhancing the skill performance and confidence of postgraduate residents.

## Supporting information

**S1 File. The multiple-choice questions.**
(DOCX)

**S2 File. The rubric for evaluating the LIO-ROP procedure.**
(DOCX)

**S3 File. The confidence assessment.**
(DOCX)

## Acknowledgments

We extend our gratitude to Ms. Sujinda Damthong and Ms. Orawan Suwannarat for their invaluable contributions to data collection and statistical analysis. Special thanks go to Dr. Alan Frederick Geater and Dr. Satid Thammasitboon for their critical review of the manuscript and their editorial guidance.

## Author contributions

**Conceptualization:** Narisa Rattanalert, Supaporn Tengtrisorn, Phanthipha Wongwai, Atchareeya Wiwatwongwana, Penny Singha, Sirinya Suwannaraj, Warachaya Phanphruk.

**Data curation:** Parichat Damthongsuk.

**Formal analysis:** Narisa Rattanalert, Supaporn Tengtrisorn, Phanthipha Wongwai, Atchareeya Wiwatwongwana, Penny Singha, Sirinya Suwannaraj, Thunyaluck Jiwanarom, Warachaya Phanphruk, Parichat Damthongsuk, Dorene F. Balmer.

**Funding acquisition:** Supaporn Tengtrisorn.

**Investigation:** Narisa Rattanalert, Supaporn Tengtrisorn, Phanthipha Wongwai, Atchareeya Wiwatwongwana, Penny Singha, Sirinya Suwannaraj, Thunyaluck Jiwanarom, Warachaya Phanphruk.

**Methodology:** Narisa Rattanalert, Supaporn Tengtrisorn, Phanthipha Wongwai, Atchareeya Wiwatwongwana, Penny Singha, Sirinya Suwannaraj, Thunyaluck Jiwanarom, Warachaya Phanphruk, Parichat Damthongsuk, Dorene F. Balmer.

**Project administration:** Supaporn Tengtrisorn, Parichat Damthongsuk.

**Supervision:** Supaporn Tengtrisorn, Dorene F. Balmer.

**Validation:** Narisa Rattanalert, Supaporn Tengtrisorn, Penny Singha.

**Visualization:** Supaporn Tengtrisorn, Dorene F. Balmer.

**Writing – original draft:** Narisa Rattanalert, Supaporn Tengtrisorn.

**Writing – review & editing:** Narisa Rattanalert, Supaporn Tengtrisorn, Phanthipha Wongwai, Atchareeya Wiwatwongwana, Penny Singha, Sirinya Suwannaraj, Thunyaluck Jiwanarom, Warachaya Phanphruk, Parichat Damthongsuk, Dorene F. Balmer.

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
