## [Decision Letter · Decision Letter 0]

8 Apr 2025

Deliberate practice for retinopathy of prematurity: Retinal laser training using schematic eyes in ophthalmology education

PONE-D-25-08829

Dear Dr.  Rattanalert

We’re pleased to inform you that your manuscript has been judged scientifically suitable for publication and will be formally accepted for publication once it meets all outstanding technical requirements.

Kind regards,

Anandhi Upendran, Ph.D

Academic Editor

PLOS ONE

Additional Editor Comments (optional):

Reviewers' comments:

Reviewer's Responses to Questions

**Comments to the Author**

1. Is the manuscript technically sound, and do the data support the conclusions?

Reviewer #1: Yes

2. Has the statistical analysis been performed appropriately and rigorously? 

Reviewer #1: Yes

3. Have the authors made all data underlying the findings in their manuscript fully available?

Reviewer #1: Yes

4. Is the manuscript presented in an intelligible fashion and written in standard English?

Reviewer #1: Yes

5. Review Comments to the Author

Reviewer #1: In the current work authors have investigated the impact of multifaceted simulated training on the confidence and skillset of residents and fellows to be able to perform laser indirect ophthalmoscopy for retinopathy

of prematurity. The study clearly shows that the training has improved the confidence and skill of participants, thereby recommending such training and facilities in all ROP prelevant countries. The manuscript is well written. Statistical nalaysis is performed methodically and the data supports the conclusions. The manuscript is appropriate for publication.

6. PLOS authors have the option to publish the peer review history of their article (what does this mean? ). If published, this will include your full peer review and any attached files.

**Do you want your identity to be public for this peer review?** For information about this choice, including consent withdrawal, please see our Privacy Policy .

Reviewer #1: No

---

## [Editor Report · Acceptance letter]

PONE-D-25-08829

PLOS ONE

Dear Dr. Rattanalert,

I'm pleased to inform you that your manuscript has been deemed suitable for publication in PLOS ONE. Congratulations! Your manuscript is now being handed over to our production team.

Kind regards,

on behalf of

Dr. Anandhi Upendran

Academic Editor

PLOS ONE